# Analysis of raindrop size distribution from the double moment cloud microphysics scheme for monsoon over a tropical station

K S Apsara[1,2], Aravindakshan Jayakumar[1], Theethai Jacob Anurose[1], Saji Mohandas[1], Paul R Field[3], Thara Prabhakaran[4], Mahen Konwar[4], and Vijapurap Srinivasa Prasad[1]

[1]National Centre for Medium-Range Weather Forecasting, Noida, India
[2]Indian Institute of Science Education and Research, Tirupati, India
[3]Met Office Exeter UK
[4]Indian Institute of Tropical Meteorology, Pune, India

**Correspondence:** K S Apsara (ksapsara123@gmail.com)

**Abstract.** Accurate precipitation forecasting hinges on the representation of microphysical processes within numerical models. A key approach to understanding these processes is through the analysis of hydrometeor drop size distribution (DSD). The characteristics of DSD bulk parameters:-Mass Weighted Mean Diameter (Dm) and the Normalized Intercept Parameter (Nw), are estimated from the double moment cloud microphysical scheme (CASIM: Cloud-Aerosol Interacting Microphysics) employed in the operational convection permitted model of National Centre for Medium-Range Weather Forecasting (NCUM-R). The observations from the Joss-Valdvogel Disdrometer (JWD) and the Global Precipitation Mission - Dual Frequency Precipitation Radar (GPM-DPR) are analyzed for providing essential validation. An algorithm for separating the monsoon precipitation into convective and stratiform types in NCUM-R and a new parameter estimation module to obtain DSD parameters from the CASIM are established in the study. The model exhibits agreement with the characteristics of the DSD of raindrops with Dm ranging from 0.5mm to 2.5 mm marking the majority of the monsoon precipitation events. However, the underestimation when it comes to the larger drops (with Dm > 3.25 mm and Rainrate >= 8 mm/hr) demands a reassessment in microphysical parameterizations. The advanced autoconversion parameterization scheme applied in CASIM favored the growth of large drops compared to the existing scheme. The enhanced growth of larger drops is reflected in the increased accuracy in the prediction of extreme precipitation associated with a convective event. The current study underscores the importance of refining microphysical parameterizations to improve the accuracy of precipitation forecasts offering a pathway for enhanced model performance in future operational forecasting systems.

## 1 Introduction

Recent advancements in numerical weather forecasting have significantly enhanced prediction accuracy across various grid scales, leading to more reliable forecasts at both global and regional levels (Brunet et al., 2023). However, notable discrepancies remain in the prediction of extreme weather events, where models tend to underestimate large-scale precipitation, creating uncertainties in forecasting the intensity of certain events (Mudiar et al., 2018; Kendon et al., 2012). This issue is particularly critical for the mesoscale precipitation systems during the tropical monsoon period, which exhibits high variability in both

dynamics and occurrence. Tropical monsoon rainfall remains the active source of freshwater in the region, but it also affects as a major disaster due to resulting floods and landslides, creating havoc in life and property every year. As its intensity has become highly variable over the recent years due to climate change and other factors (Loo et al., 2015), reliable prediction and quantification of these precipitation events are essential for safeguarding life and property.

The cloud microphysics schemes in numerical models play an important role in driving the dynamics of precipitation systems. These schemes regulate all the physical processes that result in the formation and dissipation of hydrometeors (here, rain). These underlying rain microphysical processes include autoconversion, accretion, and aggregation followed by physical processes like collision-coalescence, breakup, evaporation etc (Rosenfeld and Ulbrich, 2003; Straka, 2009). The autoconversion process initiates the formation of warm rain and converts cloud droplets to raindrops. Parametrization of the autoconversion process has evolved from the simplest Kessler-type (Kessler, 1969) to advanced Sundqvist (Sundqvist, 1978), Khairotidinov and Kogan (Khairoutdinov and Kogan, 2000), Liu and Daum (Liu and Daum, 2004), etc over years. Autoconversion along with other mentioned microphysical processes regulate the available cloud liquid water in the numerical models and result in the formation of hydrometeors. Drop Size Distributions (DSD) of these hydrometeors reflect the microphysical processes in the cloud and hence analyzing DSD is widely used for reviewing the microphysical processes in the numerical models (Chen et al., 2022; Zhu et al., 2021; Yang et al., 2019).

DSD of a precipitation event gives the microstructure of the drops as the number of drops present in the unit diameter interval in the unit volume of air. The size and number of the drops dissipated during a precipitation event represent the intensity of the event (as bigger drops mostly represent heavy convective rain and small drops or drizzles are mostly from stratiform rain) and hence, studying the DSD can also help in analyzing the precipitation systems. Microphysical DSD in NWP models is represented mainly using various Bulk parameterization (BP) or Spectral Bin Parameterization (SBP) schemes (Khain et al., 2015). The SBP scheme involves explicit representation of microphysics by calculating the particle size distributions (PSD) in various bins for all hydrometeors which gives accurate parametrization that can be used to improve the BP schemes. In the case of BP, instead of the size bin differentiated PSD, its moments are considered which makes it computationally efficient compared to SBP analysis (Kessler, 1969; Zhang et al., 2022; Khain et al., 2015). The single-moment approach specifies the mass of the hydrometeors while the double-moment includes the number density along with the mass. In recent studies, three-moment schemes are also introduced by including radar reflectivity as the sixth moment (Milbrandt et al., 2021). One of the efficient methods to analyze the precipitation forecast in the NWP model is the evaluation of these DSD from model outputs. It can aid not only in analyzing the quality of the forecast but also in working towards the developmental or improvements in the microphysics. A common method of evaluation is the validation of these model DSD with reliable observations (Yang et al., 2019).

A Ground-Based instrument that provides the DSD data which investigates mainly the bulk parameters is disdrometers. There are various types of disdrometers like Parsivel disdrometer, Joss-Waldvogel disdrometer (JWD), 2D video disdrometer, etc. which are widely used to analyze the DSD characteristics. Disdrometers classify raindrops into certain diameter intervals and provide data on the number distribution of the drop in respective diameter classes for the analysis. Even though the accuracy of the disdrometer is reliable, a ground-based instrument has limitations as it covers only a small region and there is sparse

availability of data in remote areas or over water bodies. Also, it can give information on drop size only as the drop reaches the surface around the vertical resolution of a few meters. In such cases, satellite data is one of the widely dependent alternate data sources. The Global Precipitation Mission (GPM) Dual-Frequency Precipitation Radar (DPR) is one such satellite radar which is the space-borne precipitation Radar launched after the Tropical Rainfall Measuring Mission (TRMM) Precipitation Radar (PR). The TRMM PR was already providing the three-dimensional radar echo structure of rainfall and the introduction of dual frequency gives the enhanced DSD in GPM. The main advantage of using the GPM-DPR is that it gives three-dimensional DSD data for the entire tropics. Also, the GPM data products give the classification of precipitation into different rain types such as stratiform, convective, shallow, or others based on the horizontal and vertical methods (Awaka et al., 2016; Chandrasekar et al., 2014). Hence the GPM-DPR observation along with the ground-based observation can provide a rigid framework for the observational analysis of the DSD of any selected region. So current study uses this observational framework of DSD for verification of the forecast from the Cloud-Aerosol Interacting Microphysics scheme (CASIM) (Field et al., 2023) of NCMRWF operational convection permitted model at 4km (NCUM-R) (Jayakumar et al., 2021). Similar sensitivity analysis of DSD was conducted in the WRF model rainfall simulation (Yang et al., 2019; Zhu et al., 2021), but this work is the first study of DSD bulk parameters for tropical precipitation using CASIM.

This study tries to characterize the DSD of major monsoon precipitation types from double-moment CASIM microphysics employed in NCUM-R and to discuss the reliability and scope of improvements in this advanced scheme. It involves

1. Development of an algorithm for the inclusion of convective and stratiform rain during the monsoon period. (section 3.1.2)

2. Selection of case studies of stratiform-convective mixed nature from observation and NCUM-R to estimate the DSD characteristics from NCUM-R. (section 3.3)

3. Obtain DSD characteristics for the selected events and the associated warm rain microphysical processes. (section 4.2)

4. Sensitivity study with different microphysical parametrization aiming the improvements in the DSD. (section 4.4)

## 2 Instruments and Datasets

### 2.1 JOSS - WALDVOGEL Disdrometer

The drop size measurements of JWD, from Cloud Aerosol Interaction and Precipitation Enhancement Experiment (CAIPEEX) (Prabhakaran et al., 2023) at the region of Tuljapur (18.0087° N, 76.0709° E), a rain shadow region of Western Ghats, in Maharashtra (Konwar et al., 2022; Raut et al., 2021), are used for this study (figure 1b). The JWD is an impact-type disdrometer that gives the drop size distribution (DSD) for every one minute for diameters 0.3 to 5.3 mm in 20-diameter classes (Joss and Waldvogel, 1967). The JWD data is widely used for the validation of both ground and satellite radars and for NWP models (Tokay et al., 2003; Konwar et al., 2022; Zhu et al., 2021). The DSD is mainly represented using parameters like Mass weighted Mean Diameter (Dm) and normalized number concentration parameter which is called Normalized Intercept Parameter (Nw)

which is calculated from the number of drops falling in each drop size interval as explained in section 3.2. One of the main drawbacks of the JWD is the underestimation of smaller drops during heavy rainfall due to disdrometer dead time, where two or more smaller drops fall at the same time which is considered as a bigger drop by the instrument (Tokay et al., 2003, 2005). Other possible errors are discussed in studies like Tokay et al. (2001); Konwar et al. (2022). The JWD data covering the monsoon period (June -September) of the year 2022 is used for the study.

## 2.2 Global Precipitation Mission

GPM-DPR level 2 product of Version 7 is used for DSD data in the current study. GPM-DPR has dual frequencies, Ka-band (35.5 GHz) and Ku-band (13.6 GHz) (Iguchi, 2020) which provide enhanced DSD characteristics using non-Rayleigh scattering effects. GPM-DPR Ku band (2AKu) FS data, which has 49 footprints in a scan, the footprint size is about 5 km in diameter, and has a scan swath of 245 km, is used for the study. The DSD parameters (Dm and Nw) are available in 176 height bins where the lowest clutter-free bin data is used for surface DSD analysis. The 'Dm', 'Nw', and 'Near-surface rain rate' from the Solver Module, 'Precipitation type' from the Classification Module, and related flags for years 2019, 2020, 2021, and 2022 are used for the comparative analysis in the study. The sampling area for GPM DPR includes the square grid of dimension 1 degree considered around the location of the Tuljapur, JWD region. The daily accumulated precipitation data from Integrated Multi-satellitE Retrievals for GPM (IMERG) is used in this study for selecting the precipitation events (section 3.3).

## 2.3 NCUM-R

NCUM-R (NCMRWF Unified Regional Model) is based on the regional configuration of Met Office's Unified Model (Me-tUM) and is operationally integrated twice per day for 75 hours, initialized at 00UTC and 12 UTC. The initial condition is from the operational NCMRWF global model (NCUM-G) analysis, and the lateral boundary conditions (LBC) are provided every 3 hours up to 75 hours. The science configuration for the regional model is based on Regional Atmosphere and Land version 3.1 (RAL3.1). CASIM is a multi-moment bulk microphysics scheme designed to simulate aerosol cloud interactions (Field et al., 2023). CASIM represents cloud processes by using 5 cloud species - cloud liquid, rain, ice, snow, and graupel (Miltenberger et al., 2018). For RAL3, CASIM considered aerosol fields for the droplet activation following Abdul-Razzak and Ghan (2000), while the cloud ice concentration followed the Braham et al. (1986) temperature relation. Climatology gives masses of 5 different aerosol species, which are diagnosed to create a single aerosol number. Then inversion of the number-mass relationship retrieves a single aerosol mass. The hydrometeor size distribution for each category is described by a gamma distribution for which the two moments are prognostic (related to the mass and number mixing ratios) for the five species mentioned above. In addition, fixed densities, diameter–mass relations, and diameter–fall-speed relations are assumed for each hydrometeor category. Graupel is treated similarly to the other hydrometeors and is produced through the freezing of raindrops and rain-collecting ice. More details of the represented transfer rates between different hydrometeor categories are discussed in Field et al. (2023). The performance of CASIM was found to be better than the single-moment microphysics scheme for the tropical settings.

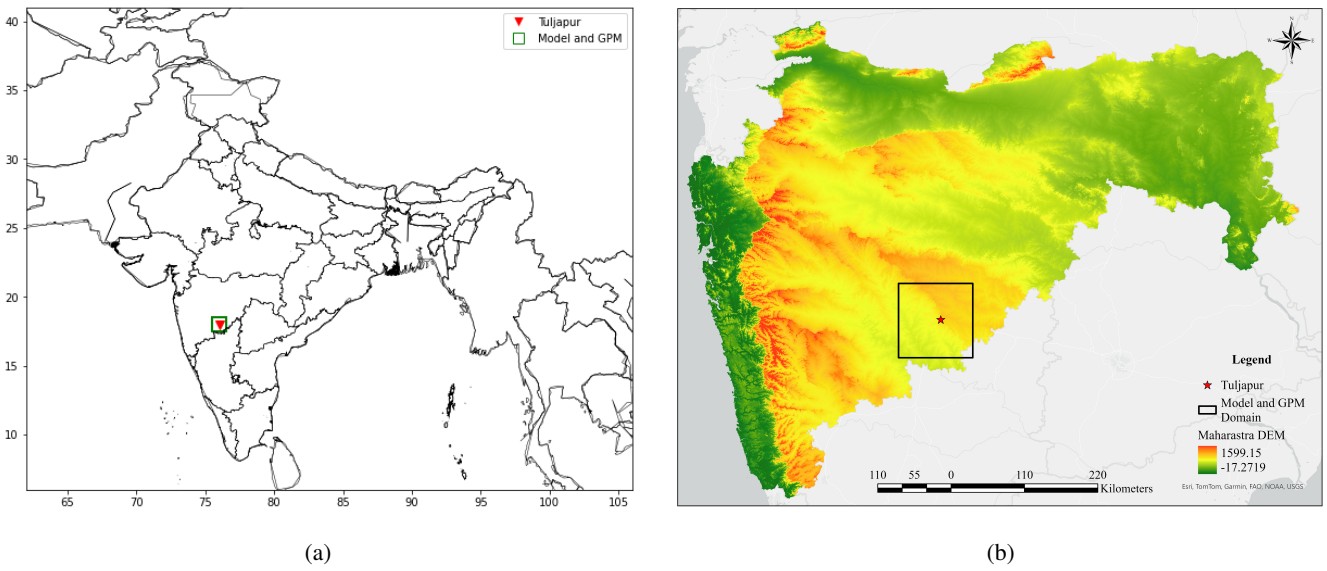

|         |         |
|---------|---------|
| (a)     | (b)     |

**Figure 1.** The location of data sampling of JWD, GPM, and Model is highlighted within the NCUM-R domain (a) and in topographical map of Maharastra (b)

A square grid formed by 75.5 E, 76.5 E, 17.5 N, and 18.5 N around the JWD location is considered for data sampling in the model the same as for GPM (figure 1).

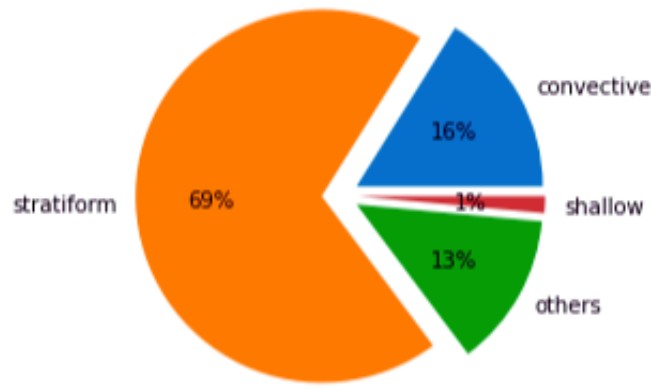

**Figure 2.** The rain type occurrence in Tuljapur (Maharashtra) ( Source: GPM-DPR)

## 3 Methodology

Before analyzing DSD characteristics, The primary goal is to discuss the algorithm for the precipitation classification in ob-
servations and NCUM-R. DSD and rain rate widely vary in stratiform and convective rain (Konwar et al., 2022). Hence
classification and segregation of precipitation types are important as they aid the analysis and also in validation of the distinct
pattern in DSD. A new algorithm for the segregation of precipitation into convective, stratiform, and others (which do not
satisfy convective or stratiform criteria) in NCUM-R is introduced in this study. The representation of DSD bulk parameters
for both observation and model are also discussed in the methodology.

### 3.1 Precipitation Classification in Observations and Model

#### 3.1.1 JWD and GPM

The JWD data is classified into different rain types, convective, stratiform, or others, by using the combined method of Bringi
et al. (2003) and the general reflectivity thresholds. The reflectivity thresholds used the surface radar reflectivity factor greater
than 35 dBZ to be classified as convective rain, to obtain the strong convective events, and to be consistent with the algorithm
used in the model for the rain type classification (section 3.1.2). The rain events that satisfy the rain intensity criteria as per
Bringi et al. (2003) for convective/stratiform along with the reflectivity thresholds of greater than or equal to 35 dBZ/less than
35 dBZ are considered to be convective/stratiform accordingly. The GPM-DPR classified the rain type such as convective,
stratiform, shallow, or others based on the combination of the horizontal method which uses the radar reflectivity thresholds,
and the vertical method, which examines the presence of bright band (Awaka et al., 2016). The rain-type distribution of Tuljapur
from GPM-DPR is shown in 2 which represents the predominant occurrence of stratiform precipitation over other rain types
over the period JJAS of 2022.

#### 3.1.2 Convection permitting model (NCUM-R)

The study proposes an algorithm to classify the model data into major precipitation types - convective, stratiform, or others
specific to monsoon. The study used a combination of dynamic conditions and rain rate thresholds to frame the algorithm. The
algorithm uses the in-cloud vertical velocity criteria as the initial basis of classification of the precipitation. Convective precip-
itation is marked by intense convection where the vertical updrafts are higher compared to the fall velocity of the hydrometeors
in the air parcel. Here the in-cloud vertical velocity (marking convection) is the sum of the vertical component of wind and the
square root of turbulent kinetic energy (tke).

The in-cloud vertical velocity ($W_c$) is calculated as:

$$W_c = w + c\sqrt{tke} \tag{1}$$

where w is the vertical component of the wind, and c is the tuning parameter with value 1. The vertical column corresponding

to each grid scale is considered a convective column if it satisfies the condition of higher updrafts in the column compared to

fall velocity. Whereas in stratiform rain the in-cloud vertical velocity is less compared to the fall velocity (Houze Jr, 1997). Additionally, small pockets of higher updraft regions can be present in the stratiform air parcel. To filter out these, the higher fall velocity condition at a minimum of three model levels up to an altitude of 15 km is considered to ensure the presence of stratiform rain in the column.

Precipitation can also be a transition type that can be neither convective nor stratiform (shallow-convective or mixed rain types). To separate this from convective or stratiform rain, rain rate thresholds at the surface levels are used. Initially, all rain pixels with rain rates greater than 55 mm/hr are considered convective. This threshold is based on the horizontal criteria used for rain-type classification (Awaka et al., 2016). Then for convective classified rainy columns, the minimum rain rate threshold of 5 mm/hr or above at the surface is considered as convective rain and those that do not satisfy the criteria are classified to

the 'other' category. Similarly, for the stratiform classified rainy column, those that satisfied the minimum threshold of 0.5 mm/hr or above at the surface are classified as stratiform rain and the rest are classified as 'other'. The detailed flowchart of the algorithm is shown in Figure 3.

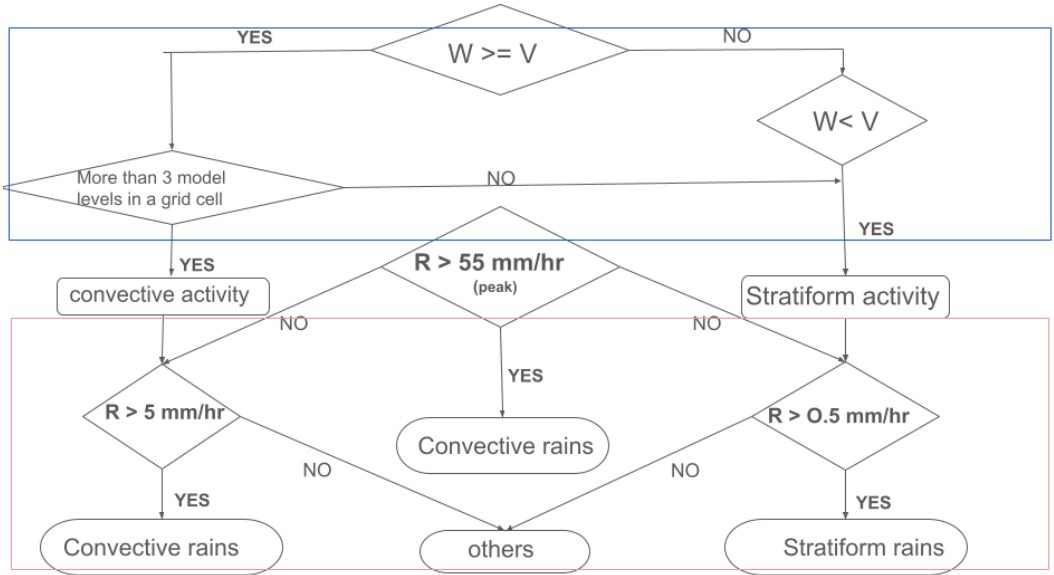

**Figure 3.** Flowchart of the algorithm used for precipitation type classification in the NCUM-R

### 3.2    Gamma distribution parameterizations from Observation and CASIM

Since the evaluation of DSD characteristics from the CASIM is a novel study, one of the important methodologies is the

incorporation of DSD diagnostics into the model to extract the desired bulk parameters in each timestep. Both observation and

model use the gamma distribution function for DSD evaluation. The DSD parameters can be calculated using the different moments or combinations of moments of the gamma distribution (Ulbrich, 1983). The generalized form of gamma distribution where the number of drops in unit diameter interval in a unit volume of air is represented by N(D) which is formulated as:

$$N(D) = N_0 D^\mu \exp(-\lambda D) \tag{2}$$


where D is the diameter where $0 < D < Dmax$, $N_0$ is the intercept parameter, $\mu$ is the shape parameter and $\lambda$ is the slope parameter.

Here the unit of $N_0$ is $m^{-1-\mu}$ which is affected by strong variation in the change of $N_0$ with respect to the $\mu$ (Chandrasekar and Bringi, 1987). To overcome this and to remove the constraint of data analysis based on shape, the normalization technique

is used following Testud et al. (2001). The normalization technique used is a mathematical technique that will allow the evaluation of the raindrop spectra and their properties based on the integral parameters of the DSDs. These integral parameters are represented by the moments of the gamma distribution.

The normalized form of the gamma DSD is represented by as:

$$N(D) = N_w f_\mu(D/D_m) \tag{3}$$


where $N_w$ is the normalized intercept parameter or normalized number concentration parameter with unit $mm^{-1}m^{-3}$, $D_m$ is mass-weighted mean diameter with unit mm. The $f_\mu(D/D_m)$ is represented by :

$$f_\mu(D/D_m) = \frac{\Gamma(4) \cdot (4+\mu)^{4+\mu}}{4^4 \cdot \Gamma(4+u)} \cdot (D/D_m)^\mu \cdot \exp\left(-(4+\mu) \cdot D/D_m\right) \tag{4}$$


$D_m$, Liquid Water Content(LWC) with unit $gm^{-3}$ and $N_w$ can be directly calculated from the moments of the gamma distribution.

The $i$th moment of DSD is defined by:

$$M_i = \int_0^\infty D^i N(D) dD \tag{5}$$

Dm calculated using the moment equations:

$$D_m = \frac{M_4}{M_3} \tag{6}$$

expanding using equation 5,

$$D_m = \frac{\int_0^\infty D^4 N(D) \, dD}{\int_0^\infty D^3 N(D) \, dD} \tag{7}$$

The LWC is proportional to the third moment $M_3$ which is represented as:

$$LWC = \frac{\Pi \cdot \rho_w \cdot M_3}{10^3 \cdot 6} \tag{8}$$

where $\rho_w$ is the density of water in $gcm^{-3}$ .

expanding using equation 5:

$$LWC = \frac{10^{-3}\pi}{6}\rho_w \int\limits_0^\infty D^3 N(D)\, dD \tag{9}$$

$N_w$ is directly proportional to LWC and inversely proportional to the fourth power of $D_m$

$$N_w = \frac{256 \cdot 10^3 \cdot LWC}{\pi \cdot \rho_w \cdot D_m^4} \tag{10}$$

In JWD the N(D) for the 20-diameter class is measured for every minute. So the bulk parameters $D_m$, $N_w$, and LWC are calculated directly using the above equation. In the case of GPM-DPR, the calculation of DSD parameters from reflectivity and attenuation constant can be found in the Iguchi et al. (2010). For double moment CASIM microphysics, the gamma distribution equation is:

$$N(D) = \frac{n_x \cdot \lambda^{1+\mu_x}}{\Gamma(1+\mu_x)} D^{\mu_x} \exp(-\lambda D) \tag{11}$$

where slope parameter $\lambda$ is

$$\lambda = \left( \frac{\pi \cdot n_x \cdot \rho_x \cdot \Gamma(4+\mu_x)}{6 \cdot q_x \cdot \Gamma(1+\mu_x)} \right)^{\frac{1}{3}} \tag{12}$$

where $\mu_x$ = shape parameter of the hydrometeor x (for rain, $\mu_x$ = 2.5), $n_x$ = number concentration of hydrometeor x in each timestep t, $\rho_x$ = density of hydrometeor x, $q_x$ = mixing ratio of hydrometeor x and $\Gamma$ = gamma function which is defined as $\Gamma(a) = \int_0^\infty e^{-t} \cdot t^{a-1} dt$.

Substituting N(D) into the equation 5, we obtain:

$$M_i = N_0 \frac{\Gamma(i+\mu+1)}{\lambda^{(i+\mu+1)}} \tag{13}$$

Where $N_0$ is represented (by comparing equation 2 to equation 11) as:

$$N_0 = \frac{n_x \cdot \lambda^{1+\mu_x}}{\Gamma(1+\mu_x)} \tag{14}$$

Following the same calculation, the DSD parameters are derived from the CASIM double moment gamma distribution model. From equation 6:

$$D_m = \frac{\Gamma(5+\mu)}{\lambda \cdot \Gamma(4+\mu)} \tag{15}$$

Similarly from equation 8 and 10:

$$N_w = \frac{256 \cdot nx \cdot \Gamma(4+\mu)}{6 \cdot \lambda^3 \cdot \Gamma(1+\mu) \cdot D_m^4} \tag{16}$$

The new diagnostics module is introduced in the CASIM where the derived equations for the DSD parameters (Dm and Nw) are added. Details of the newly added module and the call tree are in appendix A.

One of the primary limitations of the JWD is its inability to detect raindrops smaller than a certain threshold (typically Dcut≈0.3mm). To assess the potential impact of this limitation on our analysis, we conducted a detailed evaluation, as described in appendix C. It is found that the impact of this truncation on key integral parameters, such as the Dm and Nw, is minimal. This is because both the third and fourth moments of the drop size distribution (used to compute Dm) are affected proportionally by truncation, leaving their ratio largely unchanged. In our analysis, we quantify this bias using the gamma distribution formulation of CASIM PSD, applying the regularized incomplete gamma function to derive both truncated and full-distribution Dm for fixed $\lambda$ and $\mu$. Our results show that the ratio Dm(cut)/Dm(full) deviates by less than 5% for Dm >0.75, indicating that truncation introduces negligible bias in the regime most relevant to our study. Similarly as Nw is inversely proportional to the fourth power of Dm (equation 16), the impact of truncation also affect Nw but is negligible in the context of the current study. Accordingly, the shape parameter $\mu$ remains reliable, and the derived conclusions about the drop size study and microphysical characteristics are robust against instrument limitations.

### 3.3 Selection of case studies

Apart from considering the entire JJAS as in observation, certain case studies with varied contributions of convective and stratiform precipitation are selected for further model evaluation. The JWD rain amount data is accumulated for 24 hours to be consistent with the daily accumulated precipitation data from IMERG and is used to select the case dates. The JWD data is segregated into convective and stratiform based on the criteria explained in the methodology (3.1.1) and is utilized to select the precipitation events of JJAS 2022 having both stratiform and convective contributions in varied amounts. Figure 4 is the time series of daily accumulated precipitation of July 2022 from IMERG (red curve) (Thakur et al., 2020) with JWD (dashed curve) along with convective (green curve)-stratiform (blue curve) segregation. The stratiform rain is found to be prevalent in most of the days contributing to the majority of the rain type. The convective activity is most notable on 30th July as inferred from Figure 4. Generally, the IMERG data is found to agree with JWD during earlier periods of monsoon months. This relation is weakened as the monsoon recedes (Murali Krishna et al., 2017). A more detailed analysis of this data product with respect to JWD is out of the scope of the current study. The total case dates chosen from the 2022 monsoon for this study are shown in table1.

| JUNE | JULY | AUGUST | SEPTEMBER |
|---|---|---|---|
| 27-06-2022 | 08-07-2022 | 04-08-2022 | 07-09-2022 |
|  | 09-07-2022 | 13-08-2022 | 09-09-2022 |
|  | 13-07-2022 |  |  |
|  | 17-07-2022 |  |  |
|  | 30-07-2022 |  |  |

**Table 1.** case dates of selected events for simulation in NCUM-R - JJAS 2022

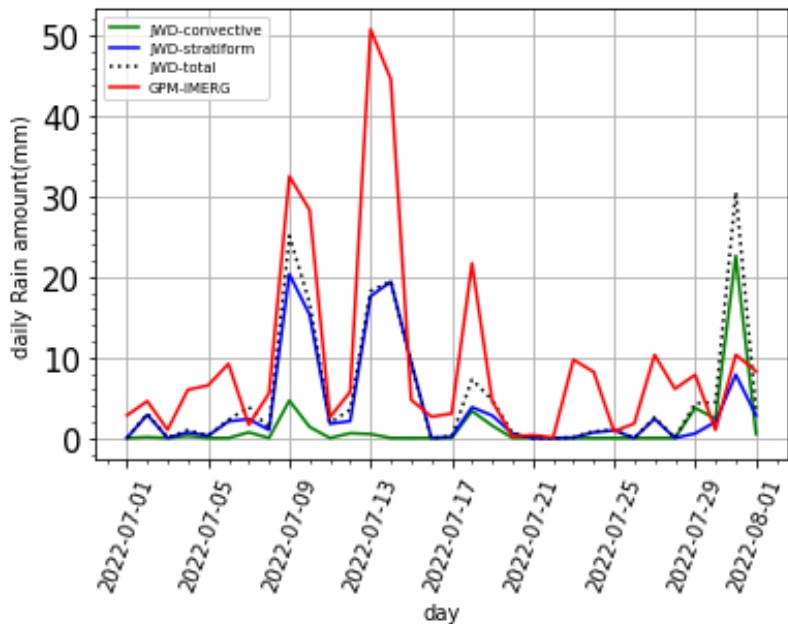

**Figure 4.** Time series plot of IMERG and JWD classified to convective and stratiform showing the case events of 2022 July

The NCUM-R version 12.0 with science configuration RAL 3.1 is used for the model simulations required for the study. Convective parametrization is switched off in the model version used and the dynamic feedback to the convection is affected by the microphysics species transfer processes. Additionally, Radar Hole correction is added to this version which happens due to the intense rain collecting rain process in tropical settings leading to a decrease in number concentration (Appendix B). Case dates selected from the JJAS months of 2022 are considered (table1) for numerical simulation in NCUM-R and the output is generated every ten minutes.

# 4 Results and Discussion

## 4.1 Convective/stratiform precipitation types in the NCUM-R model

To characterize the DSD nature of CASIM for different rain types within NCUM-R, as an algorithm is introduced in the section 3.1.2 for segregating the total rain into convective and stratiform type. A case event from 18 and 30th July 2022 is chosen to check the robustness of the algorithm and model performance in the extracting different type of precipitation compared to the observation. JWD rain rate data is hourly averaged to for this validation. However, since GPM does not have continuous data available for an entire day, it is not considered here. The time series of the hourly average rain rate (mm/hr) for 30th July 2022 and 18th July 2022 is shown in Figure 5 for both model and JWD. 30th July 2022 represents mostly convective events while 18th July 2022 marks stratiform events. The model can represent the precipitation types in agreement to the JWD observation although there is a mismatch in precipitation hours which can be due to differences in the horizontal sampling region. Overall, the model can extract and classify the convective and stratiform events as observation, hence the new algorithm is similarly applied for all the case dates (table 1) chosen under the monsoon season for further analysis.

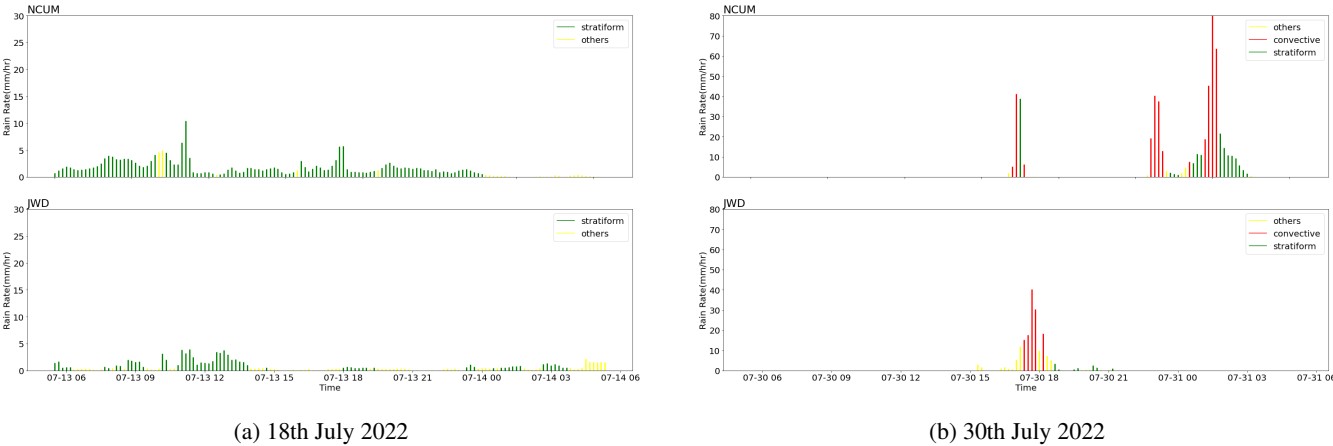

(a) 18th July 2022          (b) 30th July 2022

**Figure 5.** The time series of hourly precipitation activity of 18th July 2022 and 30th July 2022 from model and JWD segregated to different rain types (convective and stratiform)

## 4.2 Evaluation of CASIM DSD bulk parameters

The DSD parameters Dm and Nw obtained from the CASIM microphysics of NCUM-R are analyzed for selected case events. The sampling mismatch errors in GPM and JWD are small as reported by studies (Adirosi et al., 2021) and the nature of the DSD can give the general characteristics specific to the region and season. For more clarification and due to the availability of continuous data, the same case events (table 1) are chosen from JWD for a comprehensive comparison. Since GPM data is not continuous the selection of case events specifically is impractical but as specified, it is utilized to indicate the general characteristics of DSD for JJAS. Figure 6 shows the scattered plots of all sampling data points of JWD, GPM, and NCUM-R.

The model shows agreement with the JWD and GPM for raindrops with a maximum frequency of occurrence of Dm between 1 mm and 2 mm. The model underestimates the drops with Dm > 3.5 mm compared to observations and is unable to extract those drops.For Dm, the mean bias error (MBE) indicates that GPM (-0.1131) and model (-0.0112)underestimates Dm , relative to JWD . The mean absolute error (MAE) and root mean square error (RMSE) suggest moderate error spreads, with the model having a larger RMSE (0.7033) than GPM (0.5531). For Nw, the bias analysis shows that GPM significantly underestimates Nw (MBE = -968.91), whereas the model overestimates it (MBE = 1877.85). To investigate the truncation in drops (Dm > 3.5 mm) in terms of different rain types, the scatter plot of Nw vs Dm for different rain types for observation and model is studied (Figure 7). The segregated rain type scatter plot of JWD clearly distinguishes between the stratiform and convective rain without much overlap, while both in GPM and NCUM-R there is significant overlap in convective and stratiform rain. One reason for this can be the realistic unconstrained shape parameter ($\mu$) in JWD whereas in GPM and NCUM -R, $\mu$ of the raindrops is constrained as per the defined parametrizations. The $\mu$ is determined by the shape of the raindrops which changes with the air resistance, size, and other external factor. In the case of GPM and NCUM-R, since the parametrizations determine the DSD from the obtained data, the $\mu$ is constrained for ease of computation. Many advanced three-moment microphysics schemes consider $\mu$ as unconstrained but at the cost of expensive computation (Milbrandt et al., 2021). The visible dominance of stratiform drops is also an important inference from Figure 7 (NCUM-R) where most smaller drops are contributed from the stratiform precipitation.

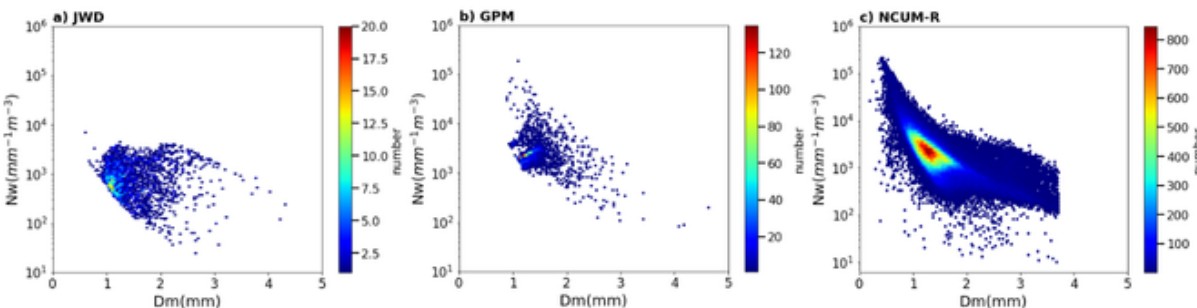

**Figure 6.** Number distribution of datapoints across Nw and Dm for observations and model

The average DSD plots for different rain rates (R)(mm/hr) intervals are in Figure 7. DSD in rain rate intervals follows gamma distribution and the model has good agreement with the observations in drops of size between 1 mm to 3 mm. The truncation of large drops ($D_m > 3.5$ mm) is visible in the figure, especially from R > 8mm/hr. The larger drops with $D_m \geq$ 3 mm is mostly are products of convective rain (Konwar et al., 2022) and hence are bound with higher rain rates. However, the model underestimates $D_m$ at higher rain rates and slightly overestimates at lower rain rates compared to JWD. The low to average-sized drops, especially drops formed during stratiform rain were shown to be almost in agreement with the model in most case events, even though there is a slight abundance of stratiform drops in $D_m<1$mm. Hence overall, NCUM-R shows more reliability in the stratiform event compared to the convective events and is unable to extract the large drops figure 7.

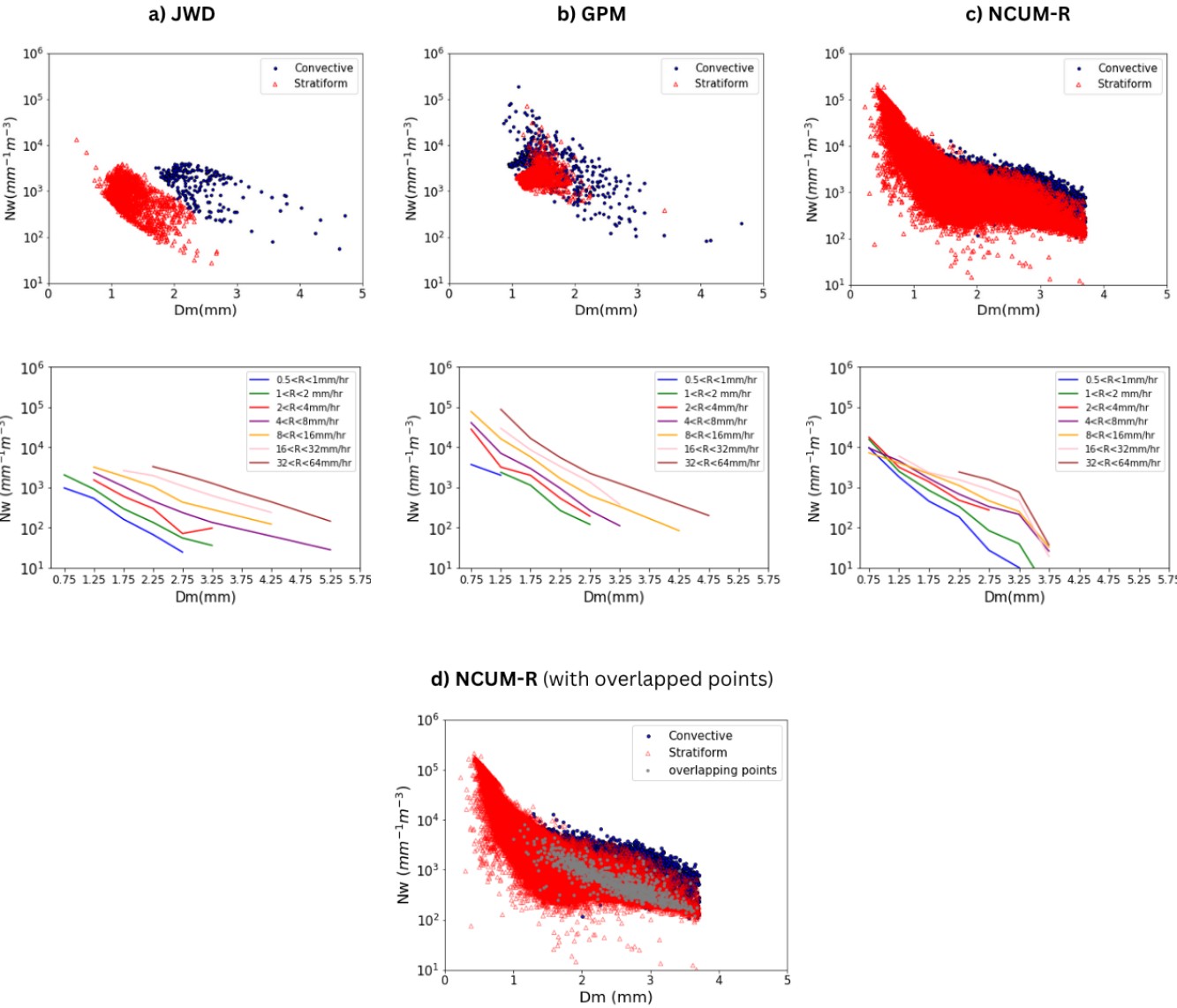

**Figure 7.** Scattered plot of Nw vs Dm segregated to different rain types(convective and stratiform)(top row) and Nw vs Dm for different Rainrates (second row) of JWD, GPM and NCUM-R (a,b and c).(d) shows the NCUM-R scattered figure with overlapping of convective and stratiform points

## 4.3 Analysis of DSD from CASIM scheme

The whole microphysical process from the formation of the cloud droplet, until it becomes any hydrometeor, is reflected in the DSD characteristics of the raindrops. In the general classification, stratiform rain is represented by rain with average rain rates and with $Dm \leq 3$ mm while convective rain forms larger raindrops ($Dm \geq 3$ mm) with higher rain rates (Konwar et al., 2022).

The main inferences from the CASIM DSD analysis include:

1. The agreement of the model for drops with size;1 mm$<D_m<$2mm, which represents the majority of concentration of drops.

2. Truncation of raindrops at a size Dm $\approx$ 3.5 mm

3. Reliable representation of stratiform rain but convective events are under-predicted.

Inferences 2 and 3 can be connected as the improper representation of bigger drops can possibly linked to the under-prediction in the convective events. The drop size is determined based on various microphysical processes underlying the formation of raindrops or other hydrometeors such as autoconversion, accretion, collision-coalescence, evaporation, etc. CASIM also uses certain thresholds to limit the size of hydrometeors which is introduced to reduce various size sorting problems. These threshold is a feature of CASIM. Currently raindrop breakup is not treated explicitly that would naturally limit how broad the DSD can become. To capture this behaviour the DSD breadth is limited using a threshold for Dm as is commonly done in other microphysics schemes (Jin et al., 2022). If the DSD breadth was unlimited then rain collecting rain can quickly remove all of the rain number concentration in deep tropical rainshafts. However, this threshold hinders the growth of a significant number of raindrops in reality and might be the cause of discrepancies seen in the DSD from NCUM-R. The thresholds may mostly represent the highly recurred values of drop size but there is a high chance that these values may vary according to the region and seasons, especially at tropics. One way of approaching this is to use more comprehensive parametrization in terms of present ones which do not fully rely on thresholds to reduce size sorting problems.

## 4.4 Sensitivity experiments using autoconversion schemes

In bulk microphysical models, the autoconversion process is a parameterized mechanism that simulates the transition of cloud water species to rainwater species due to the coalescence of cloud droplets. It represents the flux of mass and number across a size threshold, distinguishing clouds from rain particles. While this process is an idealization, it is crucial for modeling precipitation and requires careful evaluation. Currently, the autoconversion parameterization used in CASIM is the following Khairoutdinov and Kogan scheme (Khairoutdinov and Kogan, 2000) (as specified as KK00 hereafter) which is used for model simulations. In foresight of improvement of representation of deep convection in tropics, we used more comprehensive auto-conversion parameterization (Liu and Daum, 2004) (specified as LD hereafter) and introduced in place of KK00 in CASIM. LD scheme is a relative dispersion-based autoconversion scheme, which emphasizes cloud liquid water content, droplet number concentration, and relative dispersion of cloud droplets in the parametrization. The relative dispersion $\epsilon$ is taken to be 0.5 after conducting experiments using dynamic aggregation (Liu et al., 2006).

The LD autoconversion parametrization equation is represented as:
autoconversion = rate of collection of cloud droplets (P0) * threshold function(T0)

$$P_0 = 1.1 * 10^{13} \frac{(1+3\epsilon^2)(1+4\epsilon^2)(1+5\epsilon^2)q_c^2}{(1+\epsilon^2)(1+2\epsilon^2)N_c} \tag{17}$$

$$T_0 = 1/2(x_c^2 + 2x_c + 2)(1 + x_c)e^{-2x_c} \tag{18}$$

where,

$$x_c = 9.7 * 10^{-14} N_c^{\frac{-3}{2}} q_c^{-2} \tag{19}$$

here $q_c$ is the mixing ratio of the cloud droplet and $N_c$ is the number concentration of the cloud droplet (Xu et al., 2020).

The relative dispersion term in LD depends on the size of the hydrometeors (here rain) and is expected to properly segregate the drops thereby reducing the size sorting problems. In the LD scheme, the thresholds to restrict the growth are relaxed unlike the KK00 scheme, and the growth now depends mostly on the availability of the cloud liquid water. The DSD characteristics obtained from simulations of the case dates in table 1 using LD and KK00 schemes are depicted in figure 8.

### 4.4.1 Implication on DSD

The LD scheme shows more growth in the drops in $D_m > 3$mm when compared with KK00 as clearly depicted in the figure 8. Drop size dispersion-based LD scheme allows the growth and proper segregation of drops by reducing the size sorting problems as compared to KK00 as expected. This is in agreement with the Nw distribution, as towards higher rainrate there is clear narrow distribution of drops while using LD scheme, unlike the broad distribution of drops over Nw range in KK00. This denotes a more proper segregation of drops of each rainrate intervals over Dm and Nw range while using the LD scheme. Note that the concentration of larger drops are generally low as it formed by collecting the smaller drops, hence low Nw (figure 8b). This sensitivity study used the generalized parameterization of the LD scheme with approximate dispersion value. However, a more comprehensive autoconversion requires the experimental validation of the dispersion term specific to the tropical region. This can lead to the further tuning of the LD scheme and can be validated using seasonal simulations. An evaluation of a convection driven heavy precipitation events is discussed in the next section using LD scheme along with the default KK00 scheme.

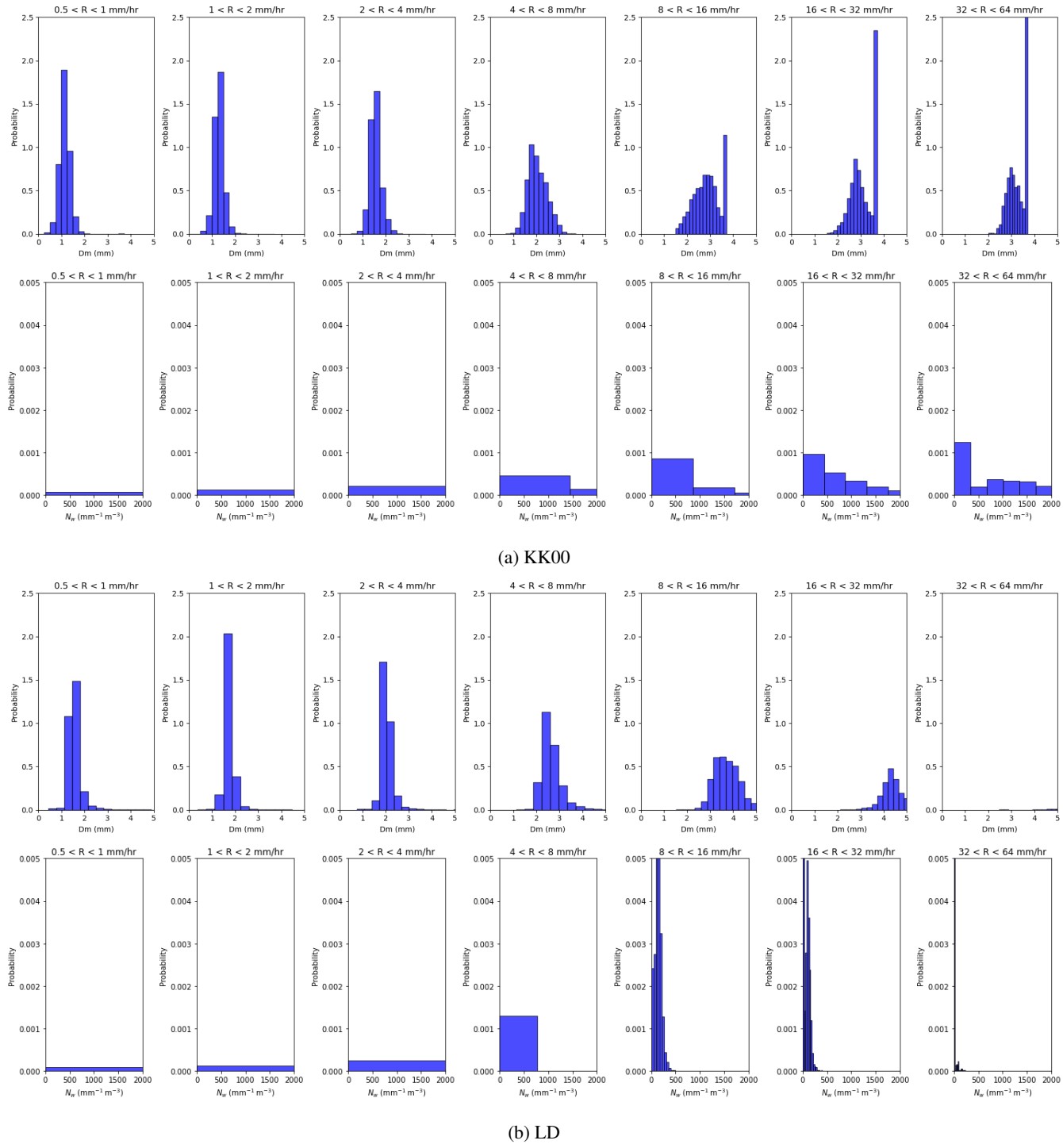

**Figure 8.** Frequency of drops corresponding to different diameters($D_m$) in each rain rate intervals

### 4.4.2 Impact on the rainfall intensity of the convective events

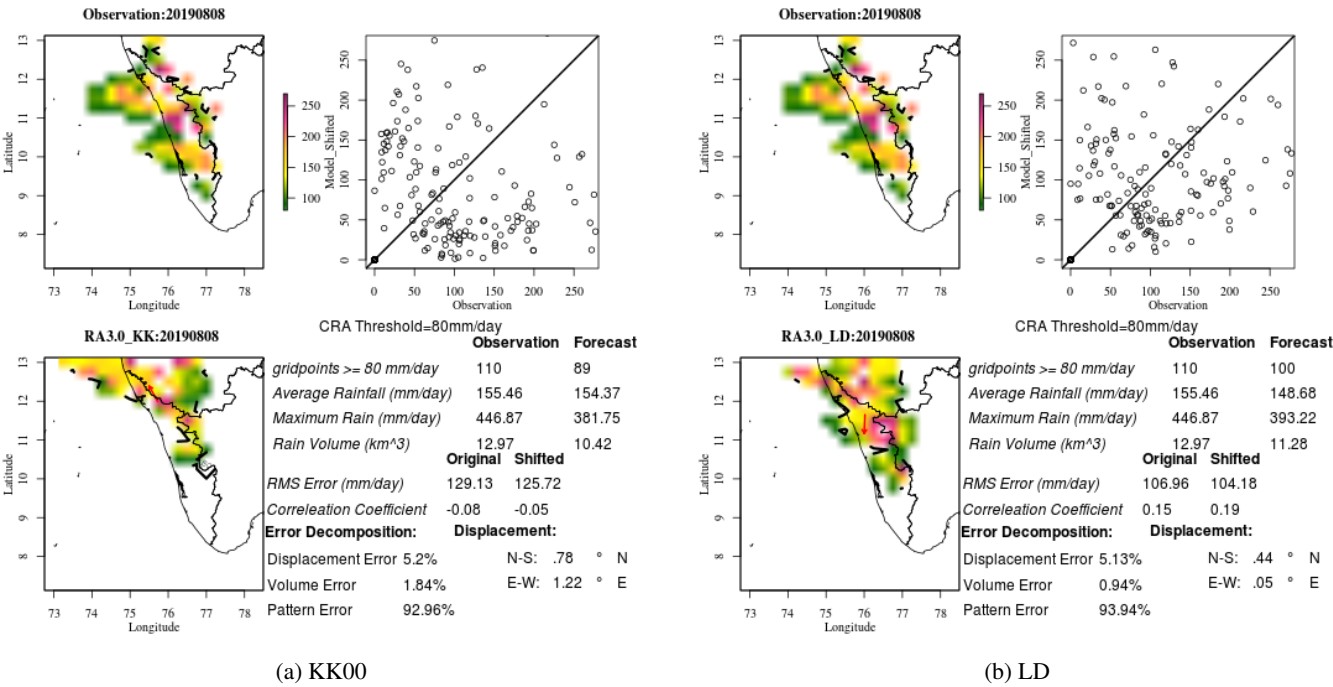

(a) KK00                                (b) LD

**Figure 9.** CRA analysis of the convective event on 8th August 2019 in comparison of observation with KK00 scheme set up(a) and LD scheme setup(b)in CASIM. The scatter plot of the observed and model grid points considered for the analysis along with statistical values of the analyzed rainfall are also shown respectively

Contiguous Rain Area analysis (CRA) is used in this study to compare and study the convective event on 8th August 2019
365   using KK00 and LD schemes. CRA analysis is a verification method to study the systematic errors in the forecast of precip-
itation events (Chen et al., 2018). It was developed by Ebert and McBride (2000) and uses the best match between forecast
and observed entities. The forecast is horizontally translated over the observations and is followed by the rotation around the
centroid of the entity (Moise and Delage, 2011). This determines the best-fit criterion as like minimum squared error (Ebert
and McBride, 2000). The total errors can be segregated as displacement error, rotation error, volume error, and pattern error.
370   In the current study, the rain rate threshold of 80 mm/hr is used and the CRA of rain rate greater than 80 mm/hr is considered
for the sensitivity study aiming at strong convective events. The current CRA analysis of the convective event on 8th August
2019 shows a reduction in the displacement (0.07 %) and volume error (0.9 %) compared to the KK00 scheme as shown in
figure 9 and table2. The lesser displacement error represents the spatial accuracy of the forecast while using the LD autocon-
version scheme for simulations. The decreased volume error on the other hand represents the amount of predicted rainfall in
the given area is more comparable to the observation while following the LD scheme. This can be due to the dependence of the
formation of raindrops on the cloud water content in the LD scheme rather than restricted by size thresholds while using the

KK00 scheme. Overall the convective event on 8th August 2019 is better represented by the LD scheme when compared with the KK00 scheme except for the pattern error (higher in LD by 0.98%).

| Error Decomposition | KK00 | LD |
|---|---|---|
| Displacement error | 5.2% | 5.13 % |
| Volume error | 1.84% | 0.94 % |
| Pattern error | 92.96% | 93.94% |

**Table 2.** Error decompositions when using KK00 and LD schemes

Note that the size of the drops is still limited in the current version of CASIM. However, based on these results of the current sensitivity tests conducted, this approach will need to be revised in the future.

## 5    Conclusions

Numerical models have shown remarkable advances in representing various mesoscale or synoptic scale atmospheric systems. Most of the advancements are from the constant developments and improvements in the mathematical parameterizations in
the numerical models representing the various atmospheric microphysical processes. Extreme events or large-scale events evidently deal with complexity in modeling due to the high variability in their dynamics. But even though these events are rarer in occurrence, they mark for large losses in life and property through intense precipitation and related disasters like floods, landslides, etc...

The tropical monsoon is a clear example that results in the formation of many dynamic systems at different scales annually.
Numerical models like NCUM-R have shown highly reliable developments in modeling tropical monsoons. One such recent development is the introduction of a double moment cloud microphysical scheme called CASIM to effectively model the various atmospheric events. So to analyze how the introduction of CASIM affects the prediction of atmospheric events of various scales, a DSD verification is conducted through the current study using JWD and GPM-DPR for observational validation. An algorithm is developed to segregate different rain types - convective and stratiform under monsoon conditions to aid the
study and it is followed by analyzing the DSD characteristics using Bulk parameters (Nw and Dm) over the Tuljapur region. The NCUM-R highly estimates all the DSD from Dm = 1 to Dm = 2.5, which marks the majority, matching most of the precipitation during tropical monsoon season. However, the drop size distribution is truncated after Dm $\approx$ 3.5 mm in CASIM, which can reflect slight discrepancies in representing large-scale or extreme events. Underestimation of the larger drops traces back to the microphysical processes in which the study concentrates on autoconversion parameterization which follows KK00
currently. A sensitivity study using the advanced LD autoconversion parameterization scheme has shown a remarkable growth of raindrops of larger diameter. This can be due to the relative dispersion-based LD scheme which utilizes the liquid water content availability to form hydrometeors. The study also analyzed how the developments in DSD impact atmospheric events. For that, a large-scale event is studied which has shown that the LD scheme in CASIM has shown lesser displacement and

volume error compared to the KK00 scheme. This resulted in a better representation of the precipitation event and hence the proper segregation of DSD can directly reflect the better prediction of precipitation events.

This study determines the improvements in DSD while using a new autoconversion scheme and validates the improvements for a selected precipitation event, however, the further validation has to be extended to more such events which can be further classified, say warm rain and cold rain, which can further provide interesting results, which is left for future studies. The threshold used for regulating the size of drops formed is relaxed in the current study to decrease its dependence on growth. However, reliable improvement demands the growth of hydrometeors that solely depend on parametrization, not using any thresholds. Further, the LD parameterization used here used relative dispersion terms which need further tuning and observational validation based on resolution and dynamic conditions. An increase in the model resolution along with the discussed parametrization modifications also lead to improvements in the forecasting capabilities.

*Data availability.*  The GPM-DPR data is available on the GES-DISC website and is accessible upon registration. JWD data is obtained from CAIPEEX team (IITM Pune) and simulation data used in the study are archived at the NCMRWF repository.

## Appendix A

### A1    Call tree for DSD diagnostics in CASIM

In CASIM, the DSD diagnostic is introduced as a new module src/dsd_cal_mod.F90. This module is linked to CASIM and UM as shown in Figure A1.

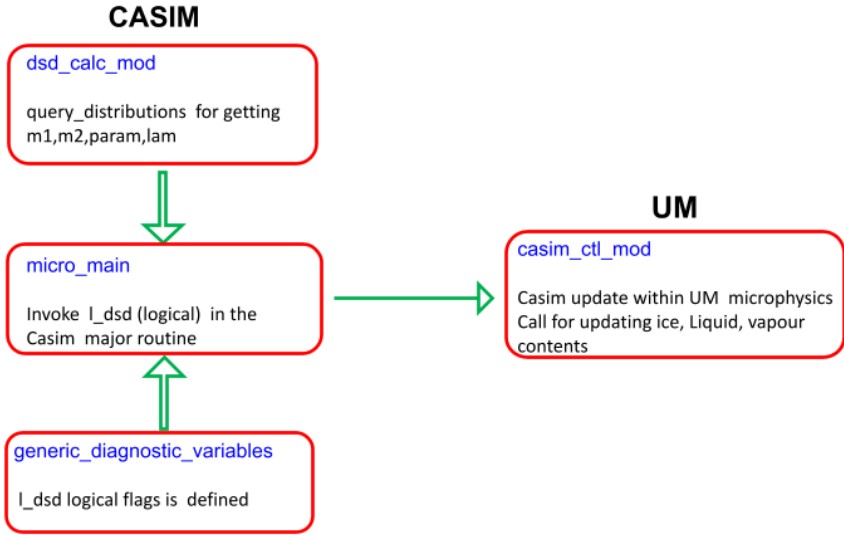

**Figure A1.** Call tree showing the necessary code modifications and its order in UM for DSD diagnostics

**Appendix B**

**B1  Radar Hole fixing bug correction**

A process of rain collecting rain which leads to decreases in the number concentration generally in intense precipitation events in tropical settings, is found in the model. When it goes below a threshold CASIM will assume nothing is present (significant mass with high rain rates, but an insignificant number) and evaporate the mass. The vapor will then condense again to form 425 clouds and evolve into rain. This is physically irrelevant and leads to holes in the radar diagnostics. To tackle this, the radar hole fixing is introduced following https://code.metoffice.gov.uk/trac/rmed/ticket/375. The radar hole process forms drops with significant mass leading to higher rain rates even in smaller drops (Figure B1). The radar hole fix corrects the DSD distribution restricting the size and number concentration of the raindrops (Figure B1).

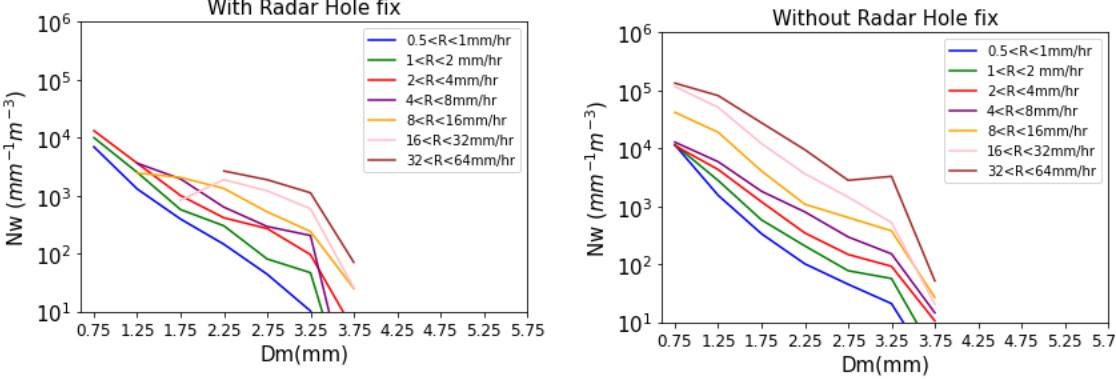

**Figure B1.** Radar hole fix correction for precipitation event on 30 th July 2022

**Appendix C**

**C1 Truncation of drops in JWD**

We have conducted an analysis to assess the impact of truncation on the mass-weighted mean diameter ($D_m$). Our approach follows the CASIM particle size distribution (PSD) framework, where $D_m$ is derived as the ratio of the 4th and 3rd moments. By utilizing the regularized incomplete gamma function, we quantify the truncation effect at $D_{cut}$, corresponding to the lowest size detected by JWD. The results indicate that while individual moments are affected by truncation, their ratio remains largely 435 unchanged. Since our histograms of $D_m$ already start above 0.5 mm, the bias introduced by truncation is minimal, and hence the shape parameter ($\mu$) remains unaffected or realistic. These findings suggest that discrepancies between JWD and model-derived $D_m$ are negligible due to truncation and the analysis discussed in the manuscript remain valid.

The CASIM particle size distribution definition is

$$N(D) = N \frac{\lambda^{1+\mu}}{\Gamma(1+\mu)} D^\mu \exp(-\lambda D) \tag{C1}$$

where $N$ is the total number concentration and $\lambda$, $\mu$ are PSD parameters.

The $p$-th moment of this distribution is given by

$$M(p) = \frac{N\Gamma(1+\mu+p)}{\lambda^p \Gamma(1+\mu)} \tag{C2}$$

The mass-weighted mean size is given by the ratio of the 4th and 3rd moments for liquid droplets:

$$D_m = \frac{M(4)}{M(3)} \qquad (C3)$$

We will use the regularized incomplete upper gamma function (`scipy.special.gammaincc`) from the Python `scipy.stats` library to estimate the moments used to compute the mass-weighted mean diameter $D_m$, assuming that $\mu$ is fixed and the same for both truncated and full distributions.

$$Q(a, x) = \frac{1}{\Gamma(a)} \int_x^\infty t^{a-1} \exp(-t)\mathrm{d}t \qquad (C4)$$

such that $Q = 1$ when $x = 0$.

For the CASIM PSD, we define $t = \lambda D$, $a = \mu + 1 + p$ where $p$ is the $p$-th moment ($p = 0$: number concentration, $p = 3$: mass concentration). The truncation is applied at $x = \lambda D_{\text{cut}}$, where $D_{\text{cut}}$ is the lowest size observed by the JWD sensor.

To estimate the ratio of JWD-derived $D_{m,\text{jwd}}$ to CASIM $D_{m,\text{mod}}$:

$$\frac{D_{m,\text{jwd}}}{D_{m,\text{mod}}} = \frac{Q(1 + \mu + 4, \lambda D_{\text{cut}})}{Q(1 + \mu + 3, \lambda D_{\text{cut}})} \qquad (C5)$$

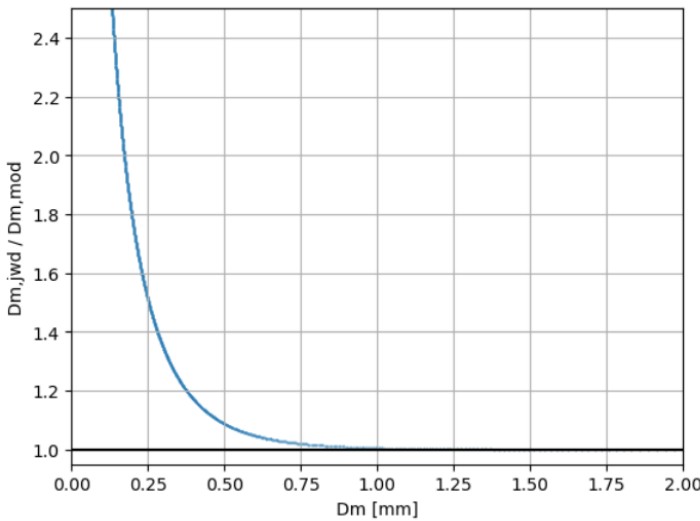

**Figure C1.** Scatter plot showing the ratio of JWD-derived mass-weighted mean diameter ($D_m$,jwd) to the CASIM model-derived mass-weighted mean diameter ($D_m$,mod) as a function of $D_m$ (in mm)

The analysis shows that the effect of truncation on the mass-weighted mean diameter ($D_m$) is minimal because both the
third and fourth moments are truncated at similar rates. As a result, their ratio remains largely unchanged. While truncation significantly affects individual moments, their proportionality ensures that $D_m$ is stable.

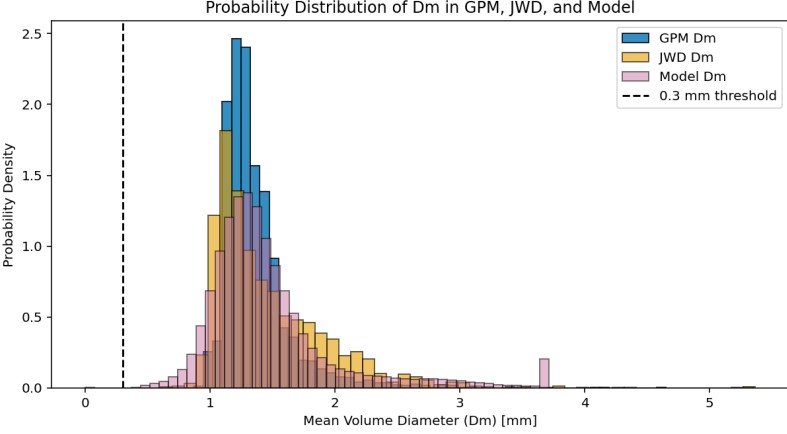

**Figure C2.** Histogram showing the probability density of drops in $D_m$ size range

Given that the histograms of $D_m$ already start at $D_m \approx 0.5$ mm, this confirms that truncation does not introduce any valid bias. Consequently, since $D_m$ remains unaffected, the shape parameter ($\mu$) is also unlikely to be impacted. Thus, truncation does not concern the validity of the results. Furthermore, because the truncation has less than a 5% effect on $D_m > 0.75$ mm, we can assert that $\mu$ should also remain unaffected.

Similary, the effect of truncation on the normalized Nw shows that the truncation leads to systematic underestimation of Nw, particularly in cases with smaller mean drop sizes ( Dm<0.5 mm). The analysis confirms that truncation effects also affect minimaly the estimation of Nw, especially for narrow and smaller drop distributions.

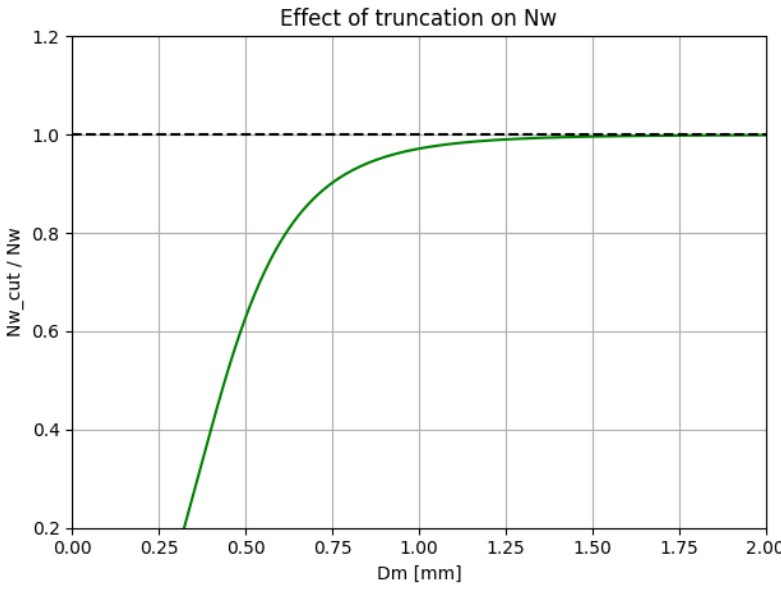

**Figure C3.** Scatter plot showing the ratio of JWD-derived Nw (Nw cut) to model-derived Nw as a function of $D_m$(in mm)

*Author contributions.* **KSA**: Conceptualization, Methodology, model simulations, post-processing, and preparation of model and observational data, analysis, writing, review and editing, original drafting

**AJ**: Conceptualization, guidance, methodology, model simulation, analysis, writing, review and editing, original drafting

**TJA**: Conceptualization, model simulations, analysis, writing, review and editing, original drafting

**SM**: guiding, methodology, model simulations, writing, review and editing, original drafting

**PF**: guiding, model simulations, writing, review and editing, original drafting

**TP**: data, writing, review and editing, original drafting

**MK**: data, writing, review and editing, original drafting

**VSP**: writing, review and editing, original drafting

*Competing interests.* The authors declare that they have no conflict of interests

*Acknowledgements.* The authors would like to express sincere gratitude to the National Centre for Medium-Range Weather Forecasting (NCMRWF), India, for providing resources and essential support for the research. Special thanks to the GPM (https://gpm.nasa.gov/data/sources/ges-disc) science team for providing the GPM-DPR dataset and to the CAIPEEX team for JWD data. The author thanks Dr. Gauri Shanker (NCMRWF) and Rose Raphel for their valuable discussions and support during this work. Lastly, we acknowledge the encouragement and constructive feedback from colleagues and reviewers, which has improved the quality of this work.

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
