# Peer review of "Analysis of raindrop size distribution from the double moment cloud microphysics scheme for monsoon over a tropical station"

_EGUsphere, 2024_

## Author Response (AR1)

**1 Comment 1**

**1.1 Comment from Referee**

From equations (12) and (15), the 0th and the 3rd moments of DSD are used to estimate Dm in this study. However, it is questionable whether JWD can accurately measure the 0th moment (number concentration of raindrops). The dead time problem and the cut-off at 0.3mm are the reasons for this. According to equation (12), underestimation of the number concentration of raindrops by JWD leads to underestimation of , which in turn leads to overestimation of Dm. It can therefore be assumed that the difference in Dm between the JWD and the simulation is due to errors in the JWD measurement. The DSDs from JWD should be corrected, for example, using the method of Raupach et al. (2019).

**1.2 Author's response**

Thank you for your comment. In response, we have conducted an analysis to assess the impact of truncation on the mass-weighted mean diameter ($D_m$). Our approach follows the CASIM particle size distribution (PSD) framework, where $D_m$ is derived as the ratio of the 4th and 3rd moments. By utilizing the regularized incomplete gamma function, we quantify the truncation effect at $D_{cut}$, corresponding to the lowest size detected by JWD. The results indicate that while individual moments are affected by truncation, their ratio remains largely unchanged. Since our histograms of $D_m$ already start above 0.5 mm, the bias introduced by truncation is minimal, and hence the shape parameter ($\mu$) remains unaffected or realistic. These findings suggest that discrepancies between JWD and model-derived $D_m$ are negligible due to truncation and the analysis discussed in the manuscript remain valid.

The CASIM particle size distribution definition is

$$N(D) = N\frac{\lambda^{1+\mu}}{\Gamma(1+\mu)}D^\mu \exp(-\lambda D) \tag{1}$$

where $N$ is the total number concentration and $\lambda$, $\mu$ are PSD parameters.

The $p$-th moment of this distribution is given by

$$M(p) = \frac{N\Gamma(1+\mu+p)}{\lambda^p\Gamma(1+\mu)} \tag{2}$$

The mass-weighted mean size is given by the ratio of the 4th and 3rd moments for liquid droplets:

$$D_m = \frac{M(4)}{M(3)} \tag{3}$$

We will use the regularized incomplete upper gamma function (`scipy.special.gammaincc`) from the Python `scipy.stats` library to estimate the moments used to compute the mass-weighted mean diameter $D_m$, assuming that $\mu$ is fixed and the same for both truncated and full distributions.

$$Q(a,x) = \frac{1}{\Gamma(a)}\int_x^\infty t^{a-1}\exp(-t)\mathrm{d}t \tag{4}$$

such that $Q = 1$ when $x = 0$.

For the CASIM PSD, we define $t = \lambda D$, $a = \mu + 1 + p$ where $p$ is the $p$-th moment ($p = 0$: number concentration, $p = 3$: mass concentration). The truncation is applied at $x = \lambda D_{\text{cut}}$, where $D_{\text{cut}}$ is the lowest size observed by the JWD sensor.

To estimate the ratio of JWD-derived $D_{m,\text{jwd}}$ to CASIM $D_{m,\text{mod}}$:

$$\frac{D_{m,\text{jwd}}}{D_{m,\text{mod}}} = \frac{Q(1+\mu+4, \lambda D_{\text{cut}})}{Q(1+\mu+3, \lambda D_{\text{cut}})} \tag{5}$$

[Figure]

Figure 1: Scatter plot showing the ratio of JWD-derived mass-weighted mean diameter ($D_m$,jwd) to the CASIM model-derived mass-weighted mean diameter ($D_m$,mod) as a function of $D_m$ (in mm)

The analysis shows that the effect of truncation on the mass-weighted mean diameter ($D_m$) is minimal because both the third and fourth moments are truncated at similar rates. As a result, their ratio remains largely unchanged. While truncation significantly affects individual moments, their proportionality ensures that $D_m$ is stable.

[Figure]

Figure 2: Histogram showing the probability density of drops in $D_m$ size range

Given that the histograms of $D_m$ already start at $D_m \approx 0.5$ mm, this confirms that truncation does not introduce any valid bias. Consequently, since $D_m$ remains unaffected, the shape parameter ($\mu$) is also unlikely to be impacted. Thus, truncation does not concern the validity of the results. Furthermore, because the truncation has less than a 5% effect on $D_m > 0.75$ mm, we can assert that $\mu$ should also remain unaffected.

**1.3 Author's changes in manuscript**

Since the authors explanation validate the methods used in the study, there is no related changes based on this comment in the manusript.

**2 Comment 2**

**2.1 Comment from Referee**

P12 Lines 267-269: The authors state that "the model shows agreement with the JWD and GPM for raindrops with a maximum frequency of occurrence of Dm between 1 mm and 2 mm", but the model seems to overestimate Nw compared to the JWD. It is desirable to have a quantitative comparison between the simulation and the observation.

**2.2 Author's response**

Thank you for your insightful comment. To address your concern, we conducted a probability density analysis, which confirmed that the maximum probability of occurrence for $D_m$ in GPM, JWD, and the model lies between 1 mm and 2 mm, as stated in the manuscript.

[Figure]

Figure 3: Histogram showing the probability density of drops in $D_m$ size range

Additionally, we performed a quantitative bias analysis to assess the differences between the simulation and observations.

For $D_m$, the mean bias error (MBE) indicates that GPM (-0.1131) and model (-0.0112)underestimates $D_m$, relative to JWD . The mean absolute error (MAE) and root mean square error (RMSE) suggest moderate error spreads, with the model having a larger RMSE (0.7033) than GPM (0.5531).

For $N_w$, the bias analysis shows that GPM significantly underestimates $N_w$(MBE = -968.91), whereas the model overestimates it (MBE = 1877.85). The high RMSE values for both GPM and the model (6013.88) suggest considerable variation in $N_w$ predictions.

The current study focuses on assessing the underestimation of larger raindrops, which, despite their lower number density, play a crucial role as they are typically associated with convective rainfall. Since convective processes contribute significantly to precipitation intensity and accumulation, understanding biases in larger drop representations is essential for improving model performance and precipitation estimates.

**2.3 Author's changes in manuscript**

The result of the quantitative bias analysis is added to the manuscript to the manuscript.

**3 Comment 3**

**3.1 Comment from Referee**

Equation (8): 103 should be in the numerator because the unit of LWC is [g/m3] and the unit of w is [kg/m3].

**3.2 Author's response**

Thank you for your comment. The discrepancy arises due to the unit system used in our derivation. We expressed $\rho_w$ in g/cm³ instead of kg/m³, which eliminates the need for the factor $10^3$ in the numerator. However, we acknowledge that we did not explicitly mention the unit conversion in the derivation, which may have caused the confusion. We will clarify this in the revised manuscript to ensure consistency in unit representation.

**3.3 Author's changes in manuscript**

The unit is specified explicitly on the manuscript.

**4  Comment 4**

**4.1  Comment from Referee**

Fig. 7c: The plots for convective precipitation are not visible due to overlap.

**4.2  Author's response**

[Figure]

Figure 4: The DSD distribution shows convective and stratiform drops and the overlapped points show the presence of both convective and stratiform drops in the region.

Thank you for your valuable feedback. We acknowledge that the overlapping points in Figure (c) are not visible. Since GPM and JWD have minimal overlap in drop size distributions, we applied the same pattern to the model for consistency in comparison. However, as you rightly pointed out, to improve visualization, a revised figure is provided below where the overlapping points are more clearly distinguishable.

**4.3  Author's changes in manuscript**

The figure to visualise the overlapping points is added to the manuscript.

**Comments from referee**

There is a basic concept error about autoconversion process. The authors state that "The autoconversion process is a primary microphysical process in which the cloud droplets collect the raindrops to form a bigger drop." This is totally wrong. According to glossary of meteorology from AMS, autoconversion means "The initial stage of the collision–coalescence process whereby cloud droplets collide and coalesce to form drizzle drops". This process doesnot directly produce big raindrop at all.

**Author's response**

We sincerely thank the reviewer for pointing out the misstatement regarding the autoconversion process. Upon review, we acknowledge that our original statement was incorrect and does not align with the widely accepted definition provided by the AMS Glossary of Meteorology.

While autoconversion is not a discrete process observed in the real-world continuum of collision-coalescence, it is a conceptual parameterization used in bulk microphysical models. In these models, autoconversion represents the flux of mass and number across a size threshold, which separates the cloud water from rainwater species. Specifically, the parameterization simulates the coalescence of small cloud droplets into drizzle or rain-sized particles, facilitating the conversion of cloud water species into rainwater species. Beyond this threshold, processes such as self-collection within the rain species further increase the mean size of raindrops.

Implications for our Study:

- ○ Autoconversion remains an essential process to test and evaluate in microphysical schemes, as it plays a critical role in the partitioning of water species and impacts precipitation characteristics in bulk models.
- ○ This process, while idealized, allows models to account for the continuum of droplet growth and serves as a proxy for the early stages of the collision-coalescence process
- ○ The evaluation and sensitivity experiments conducted in this study were not intended to enhance the growth of raindrops but rather to assess how a more advanced parameterization impacts the representation of the drop size distribution (DSD) compared to the existing approach.

**Author's changes in manuscript**

The authors revised the text in our manuscript as follows to accurately reflect the concept ; "In bulk microphysical models, the autoconversion process is a

parameterized mechanism that simulates the transition of cloud water species to rainwater species due to the coalescence of cloud droplets. It represents the flux of mass and number across a size threshold, distinguishing clouds from rain particles. While this process is an idealization, it is crucial for modeling precipitation and requires careful evaluation."

**Comments from referee**

The comparison of DSD parameters from JWD, DPR, and NCUM-R after stratiform-convective separation is even not an apple-to-apple comparison. Three separation algorithms are based on different criteria and physical concept. The same criteria should be used, such as a simple reflectivity threshold of 35 dBZ.

**Author's response**

We acknowledge the concern raised by the reviewer about the 'apple to apple' comparability of stratiform and convective separation criteria used in different datasets. While using the uniform reflectivity threshold might provide consistency, at the same time, it overlooks the unique strengths and limitations of each dataset.

1) JWD data lacks a direct calculation of reflectivity measurements; it relies on the rain rate and drop size to obtain reflectivity. So, using rain rate thresholds which is a derived parameter used to classify stratiform and convective rain, is more meaningful and sensible.

2) GPM-DPR, which has radar-based reliable capabilities, uses multiple criteria using both the presence of a bright band and the horizontal method, both evaluate the reflectivity thresholds across the vertical profiles and horizontal gradients, which is not possible to evaluate from JWD as it is a ground observation instrument. (reference: https://doi.org/10.1175/JTECH-D-16-0016.1)

3) NCUM-R is a non-hydrostatic model that uses vertical velocity as a prognostic variable, hence for tropics, it can provide a comprehensive classification of convective and stratiform rain from the embedded convective system. This along with the combined rain rate criteria marks a more reliable classification of rain. (Houze Jr, R. A.: Stratiform precipitation in regions of convection: A meteorological paradox? Bulletin of the American Meteorological Society, 78, 2179–2196, 1997.)

In terms of the above-mentioned points, using data-specific separation criteria is relevant in the context of each system as:

- ○ Each dataset has unique capabilities and limitations so using the same criteria (35dBZ) does not promise to fulfill the intention of the study.
- ○ The study tried to use the best possible method for each dataset, as the motive was to more accurately separate the convective and stratiform precipitation, hence evaluating the drop size distribution pattern represented by each dataset and analyzing it further in terms of microphysical processes.
- ○ Recent work by Peinó et al (https://doi.org/10.3390/rs16142594) used data-specific criteria for similar classifications for their objectives.

**Author's changes in manuscript**

Since the authors explanation validate the methods used in the study, there is no related changes based on this comment in the manusript.

**Comments from referee**

Figure 7: Within the Dm-Nw framework, the precipitation rate R can be directly calculated for a given shape parameter μ. The differences simply come from the sample error and truncation error, which have no physical meaning.

**Author's response**

Thank you for your valuable comment. A similar concern was raised by reviewer 1, and in response, we conducted a detailed analysis to assess the impact of truncation on the mass-weighted mean diameter (Dm). Our approach follows the CASIM particle size distribution (PSD) framework, where Dm is derived as the ratio of the fourth and third moments. Using the regularized incomplete gamma function, we quantified the truncation effect at Dcut, which corresponds to the lowest droplet size detected by JWD.

The results indicate that while truncation influences individual moments, their ratio remains largely unchanged. Given that our histograms of Dm already start above 0.5 mm, the bias introduced by truncation is less than 10%. Consequently, the shape parameter (μ) remains unaffected as Dm is realistic, and the discrepancies between JWD and model-derived Dm are not a result of truncation.

Additionally, sample errors in Dm can be related to errors in the precipitation rate (R) within the Dm-Nw framework. However, since Dm is relatively insensitive to truncation and its effect on μ is negligible, the analysis-based Dm remains valid. The detailed derivation is attached.

**Author's changes in manuscript**

Since the authors explanation validate the methods used in the study, there is no related changes based on this comment in the manusript.

---

## Referee Report (RR1)

Comments on the revised manuscript of "Analysis of raindrop size distribution from the double moment cloud microphysics scheme for monsoon over a tropical station"

General comments:

I have read responses from the authors. I appreciate the revisions, but I feel that some points are still unclear.

Specific comments:

1. Regarding the response to my first comment, the authors estimate the effect of the truncation of the raindrop size distribution as:

$$\frac{D_{m,jwd}}{D_{m,mod}} = \frac{Q(1 + \mu + 4, \lambda D_{cut})}{Q(1 + \mu + 3, \lambda D_{cut})}$$

However, it should be

$$\frac{D_{m,jwd}}{D_{m,mod}} = \frac{\lambda_{mod}}{\lambda_{jwd}} \frac{Q(1 + \mu + 4, \lambda D_{cut})}{Q(1 + \mu + 3, \lambda D_{cut})}$$

because $\lambda$ is a function of the zero-order moment and is affected by the truncation.

2. Does Figure 1 in your response show the results with a fixed $\mu$ ? Please show the results obtained by varying $\mu$ within the observed range.

3. The effect of the truncation on $N_w$ should also be assessed.

4. The influence of the truncation of raindrop diameter is an important issue that affects the conclusions of this paper. Thus, these discussions should be reflected in the main text.

---

## Author Response (AR2)

**Comment 1**

Regarding the response to my first comment, the authors estimate the effect of the truncation of the raindrop size distribution as:

$Dm,jwd \ / \ Dm,mod = Q(1+\mu+4,\lambda Dcut) \ / \ Q(1+\mu+3,\lambda Dcut)$

However, it should be

$Dm,jwd \ / \ Dm,mod = \lambda mod \ / \ \lambda jwd * Q(1+\mu+4,\lambda Dcut)/Q(1+\mu+3,\lambda Dcut)$

because λ is a function of the zero-order moment and is affected by the truncation

**Author's response**

We thank the reviewer for the insightful observation regarding the correct formulation of the impact of truncation on the mass-weighted mean diameter, Dm. We agree that λ is a function of the zeroth-order moment and is affected by truncation, and thus, a general expression comparing truncated and full distributions should account for the change in λ. However, in our specific implementation, we aimed to quantify the effect of truncation on Dm , as calculated by the JWD disdrometer, which is unable to observe drops below a certain diameter (e.g., 0.3 mm). In practice, these instruments use the same model parameters (i.e., λ and μ derived assuming a complete distribution) and apply them to a truncated drop size spectrum.

Therefore, what we aimed to assess is:

1. Calculate the true model-derived mass-weighted mean diameter $Dm_{model}$ using the full gamma distribution (from 0 to ∞) with fixed λ and μ.

2. Then, using the same λ and μ, calculate the truncated $Dm_{cut}$, by integrating only over $D>D_{cut}$, mimicking what a disdrometer would measure.

3. Finally, we compute the ratio $Dm_{cut}/Dm_{model,}$ which tells us how much the mass-weighted mean diameter is biased by truncation. This ratio can then be applied to model output to interpret what a truncated observation would report.

Comparing λ values directly between truncated and full distributions would require inversion and iteration, as λ depends on total concentration, which is affected by truncation. In light of this, we believe our current approach — fixing λ and μ and assessing the shift in Dm due to truncation — is appropriate for evaluating instrument bias (that is, simply reporting total mass for D>Dcut/total number for D>Dcut) while holding the underlying DSD fixed. We welcome further clarification from the reviewer if there are additional considerations we may have overlooked, particularly regarding an alternative approach to relate the change in λ itself through truncation.

[Figure]

**Comment 2**

**Author's Response**

2) We thank the reviewer for this important suggestion. As correctly noted, the original figure in our response showed the results for a fixed shape parameter µ=2.5.

In response to the comment, we have now extended the analysis to explore the sensitivity of the truncation effect to variations in µ, covering a realistic observed range of µ=0 to 6. The updated figure illustrates the ratio $Dm_{cut}/Dm$ as a function of Dm for each value of µ.

[Figure]

The results indicate that while the overall pattern of overestimation of Dm due to truncation remains consistent, the magnitude of this bias is dependent on the value of µ. Specifically, lower µ values (broader distributions) tend to show larger truncation-induced biases, as they contain a greater proportion of small drops that are excluded below the disdrometer threshold.

**Comment 3**

The effect of the truncation on Nw should also be assessed.

**Author's Response**

3) We thank the reviewer for pointing out the importance of evaluating the effect of truncation on the normalized intercept parameter Nw. To address this, we have carried out a numerical calculation of Nw using the gamma drop size distribution with a fixed concentration and shape parameter (μ=2.5). We compute Nw from the full DSD, and again from a truncated version of the same distribution, excluding drops below Dcut = 0.3 mm, simulating a typical disdrometer threshold. We then calculate and plot the ratio Nw$_{cut}$/Nw as a function of the true Dm. The results show that the truncation leads to systematic underestimation of Nw, particularly in cases with smaller mean drop sizes ( Dm<0.5 mm), where the bias can exceed 80%. The analysis confirms that truncation effects also affect the estimation of Nw, especially for narrow and smaller drop distributions.

[Figure]

**Comment 4**

The influence of the truncation of raindrop diameter is an important issue that affects the conclusions of this paper. Thus, these discussions should be reflected in the main text.

**Author's response**

Thanks for the suggestion, and we made an additional brief section about the truncation effect in the manuscript.

**Changes in Manuscript**

A brief paragraph about the truncation effect of JWD and how it affects our analysis is added in the manuscript's main text, and a detailed explanation about this is added as Appendix 3 in the manuscript

---

## Author Response (AR3)

**Comment 1:**

Section 4.4: The authors must explain why truncation occurs at Dm > 3.5 mm when using the Khairoutdinov and Kogan (2000) autoconversion scheme. Does the KK00 formulation include any truncation for Dm? Or, is this a problem inherent to an assumption in CASIM? As currently written, readers might believe that there is a problem in the KK00 formulation itself.

**Author's Response:**

We thank the reviewer for pointing this out. We can clarify that the KK00 scheme does not have any truncation added to it. That is as is published. CASIM does not have a raindrop breakup formulation. So to stop the rain DSD becoming too broad and reducing rain number concentration to zero through rain self collection the width of the distribution (through Dm) is limited is commonly done in other microphysics schemes (Jin et al. 2022). This is a future development area. In the current study we are experimenting with the LD scheme which gives a more realistic DSD more independently of such threshold, as explained in the manuscript. Since this statement has created slight confusion as per the comment, we have added a short paragraph to the manuscript.
Jin et al. 2022: https://www.sciencedirect.com/science/article/pii/S0169809522001314

**Changes in Manuscript:**

In section 4.3 and 4.4 we have added the following text.
"This threshold is a feature of CASIM. Currently raindrop breakup is not treated explicitly that would naturally limit how broad the DSD can become. To capture this behaviour the DSD breadth is limited using a threshold for Dm as is commonly done in other microphysics schemes (Jin et al. 2022). If the DSD breadth was unlimited then rain collecting rain can quickly remove all of the rain number concentration in deep tropical rainshafts."
"However, based on these results of the current sensitivity tests conducted, this approach will need to be revised in the future."

**Comment 2:**

Figure 8: Histograms for Nw are also needed because DSDs depend on both Nw and Dm.

**Author's response:**

We agree with the reviewer that DSD depends on both Nw and Dm, and the current version of manuscript included the growth of the drop especially the threshold part were focussed. So we have included the comparison of Nw distribution for KK00 and LD is added to the new version.

**Changes in Manuscript:**

Histograms of Nw and the following description is added to section 4.4.1 of the Manuscript.

"This is in agreement with the Nw distribution, as towards higher rainrate there is clear narrow distribution of drops while using LD scheme, unlike the broad distribution of drops over Nw range in

KK00. This denotes a more proper segregation of drops of each rainrate intervals over Dm and Nw range while using the LD scheme. Note that the concentration of larger drops is generally low as it is formed by collecting the smaller drops, hence low Nw (figure 8b)."

**Comments 3,4,5:**
(3) Line 32: "rain droplets" -> "raindrops"
(4) Line 88: "Normalized number concentration parameter" -> "normalized intercept parameter"
(5) Line 91: Tokay et al. (2005, 2003) -> Tokay et al. (2003, 2005).

**Response:**
We thank the reviewer for these specific errors . We agree with your suggestion and have replaced "normalized number concentration parameter" with the standard term "normalized intercept parameter (Nw)" in Line 88 for clarity and consistency with the literature.

**Changes in the Manuscript**
line 88, line 32 and line 91 in manuscript are corrected accordingly.

**Comment  6:**
Figure 1: Please expand the NCUM-R calculation domain and show topographical features.

**Author's Response and change in Manuscript**
Thanks for the comment and pointing out the importance of a topographical map in the study context. We have included the expanded topographical map with NCUM-R study domain and Tuljapur marked on it.

**Comment 7:**
Figure 5: The left and right panels appear to be swapped. The text states that there was more convective precipitation on July 30, but Figure 5 shows mostly stratiform precipitation.

**Author's Response:**
We thank the reviewer for identifying this and we agree that the left and right panels in Figure 5 were inadvertently swapped. We have now corrected the figure so that the panel corresponding to July 30 accurately reflects the increased convective precipitation, in line with the description in the text.

**Changes in Manuscript:**
Figure 5 has been updated to correctly reflect the intended panel assignments for July 30 and July 18.

**Comment 8**
Figure 9: The statistical values written within the figure should be listed in a separate table.

**Author's response and changes in the manuscript**
A table with the statistical values has been added in the paper.

**Comment 9:**

Figure 9: Do the individual dots in the scatter plot between observation and model_shifted represent data from the model grid points? An explanation is needed in the caption.

**Author's response:**
Thank you for your comment. We confirm that each dot in the scatter plot represents grid points where both model and observations are available.

**Changes in Manuscript**
A detailed caption is given for figure 9 as:

'CRA analysis of the convective event on 8th August 2019 in comparison of observation with KK00 scheme set up(a) and LD scheme setup(b)in CASIM. The scatter plot of the observed and model grid points considered for the analysis along with statistical values of the analyzed rainfall are also shown respectively'

---

## Author Response (AR4)

**Remarks**

Please ensure that the colour schemes used in your maps and charts allow readers with colour vision deficiencies to correctly interpret your findings. Please check your figures using the Coblis – Color Blindness Simulator (https://www.color-blindness.com/coblis-color-blindness-simulator/) and revise the colour schemes accordingly with the next file upload request. -> Fig. C2

**Author's Response**

Thank you for your suggestion regarding the color scheme accessibility in Figure C2. As advised, we have re-evaluated the figure using the Coblis – Color Blindness Simulator and identified potential issues in the red-blind and green-blind views. The figure has now been revised by adopting a colorblind-safe palette and adjusting transparency for overlapping areas.

**Changes in Manuscript**

The revised Figure C2 is included in the updated manuscript.